



# CAMELS-FR dataset: A large-sample hydroclimatic dataset for France to explore hydrological diversity and support model benchmarking

Olivier Delaigue[1], Guilherme Mendoza Guimarães[1], Pierre Brigode[1,2,3], Benoît Génot[1,4],
Charles Perrin[1], Jean-Michel Soubeyroux[5], Bruno Janet[6], Nans Addor[7,8], and Vazken Andréassian[1]

[1]Université Paris-Saclay, INRAE, HYCAR, Antony, France
[2]Université Côte d'Azur, CNRS, OCA, IRD, Géoazur, Sophia-Antipolis, France
[3]Université de Rennes, CNRS, Géosciences Rennes, Rennes, France
[4]Now at U.R.B.S., Saint-Étienne, France
[5]Météo-France, DCSC, Toulouse, France
[6]Service central d'hydrométéorologie et d'appui à la prévision des inondations (MTES/DGPR/SCHAPI), Toulouse, France
[7]Fathom, Bristol, UK
[8]Geography, University of Exeter, Exeter, UK

**Correspondence:** Olivier Delaigue (olivier.delaigue@inrae.fr)

**Abstract.** Over the last decade, large-sample approaches, i.e. based on large catchment sets, have become increasingly popular in hydrological studies. Efforts were made to assemble and disseminate national catchment datasets. This article aims to add a stone to the construction of a large international database of catchments by proposing the `CAMELS-FR dataset`, a contribution to the CAMELS initiative (Catchment attributes and meteorology for large-sample studies). The first version presented here gathers hydroclimatic data and physical attributes for a set of 654 catchments in France. These catchments cover a wide spectrum of hydroclimatic conditions (from oceanic to continental, mountainous or Mediterranean conditions), and are considered to have limited human influence. Data include time series of daily streamflow (with at least 30 years over the 1970-2021 period; also aggregated to monthly and yearly time steps) and of eleven catchment-scale daily climate variables (including precipitation, potential evaporation, and air temperature), as well as a total of 255 catchments attributes organised in ten classes (e.g. geology, soil, land cover, etc.). River flow time series were quality-checked. Along with the database itself, two graphical tools are proposed, namely dynamic graphs to visualize time series and graphical fact sheets to summarize the main catchments characteristics. Care was taken to provide as many metadata as possible to help users interpret their results based on this dataset. We intend to update the database regularly to include new available data and account for end-users' feedbacks.

## 1 Introduction

In the early days of hydrological modelling, access to data, followed by computing resources, was a major limiting factor of progress, and many of the scientific articles of that time reported results obtained on a single catchment (and sometimes using only a few flood events). The precursors of modern hydrological models were conscious that this was a major shortcoming. Linsley for example did recommend (1982, p. 15) that "because almost any model with sufficient free parameters can yield





good results when applied to a short sample from a single catchment, effective testing requires that models be tried on many
catchments of widely differing characteristics, and that each trial cover a period of many years." Roughly 20 years later, under
the tireless leadership of John Schaake, the so-called *MOPEX* dataset started to be used worldwide (Schaake et al., 2006),
blowing a refreshing wind on the hard-drives of data-thirsty modellers. Ten years later, the paper published by Gupta et al.
(2014) presented an already impressive list of 94 "large-sample" studies focusing on rainfall-runoff modelling, which had used
a dataset of more than 30 catchments. Today, the number of such studies would probably be multiplied by 10, and it has
become almost impossible to keep track of all of them. Obviously, computing time difficulties and data sharing possibilities
have completely changed.

In France, there has been a long tradition of working with large datasets, following the recommendations of two French
leading hydrologists of the 20$^{th}$ century: Maurice Pardé and Marcel Roche. Later however, large datasets were sometimes
mocked as "hydro-bulimia" (Andréassian et al., 2009), and during a certain time, the academic world favoured the production
of individual basin monographs rather than modelling studies involving large datasets. Over the last decade, progresses were
made towards automatizing the production (and the updating) of large catchment datasets. In collaboration with the main data
producers of hydrometric and meteorological data, we worked to produce the reference hydrological dataset that we present
in this paper. For the selection of this dataset, we used what we considered to be "high" quality standards, and because we
do acknowledge that there is a part of subjectivity in this, let us paraphrase Ghislain de Marsily in his comment on model
validation and state that we strove to do "our level best" (de Marsily et al., 1992), using a variety of automatic data verification
procedures, which were complemented by manual checks (including a time-consuming —but clearly necessary— visual data
inspection).

With this work, we wish to contribute to the general effort to provide large hyrdoclimatic datasets as it was done in the United
States of America (Newman et al., 2015; Addor et al., 2017), Canada (Arsenault et al., 2016), Chile (Alvarez-Garreton et al.,
2018), Great Britain (Coxon et al., 2020), Brazil (Chagas et al., 2020; Almagro et al., 2021), Australia (Fowler et al., 2021),
Central Europe (Klingler et al., 2021), Africa (Tramblay et al., 2021), Denmark (Koch, 2021; Liu et al., 2024), Switzerland
(Höge et al., 2023), Saxony (Hauffe et al., 2023), Germany (Loritz et al., 2024), and next to the many others that will follow.

Before moving to the presentation of the dataset, let us warn the reader that for reasons of scientific ethics, we refused to
exclude hydrological "outliers" (i.e. catchments that exhibit unusual hydrological behavior) from the `CAMELS-FR dataset`.
We did so because we believe that the outliers are part of the "natural hydrological diversity" (Andréassian et al., 2010), and also
because from a scientific point of view, even good-looking catchments can turn into a "modeller's nightmare" (Refsgaard and
Hansen, 2010). This is why we did keep for example the karstic catchments which have been excluded from other datasets. Note
that other catchment sets at the national scale already exist in France for more specific purposes, for example the Reference Low
Flow Network to study the long-term evolution of low flows (see Giuntoli et al., 2013; Hodgkins et al., 2024), and elsewhere
(see a review of large-sample studies in  Gupta et al., 2014).



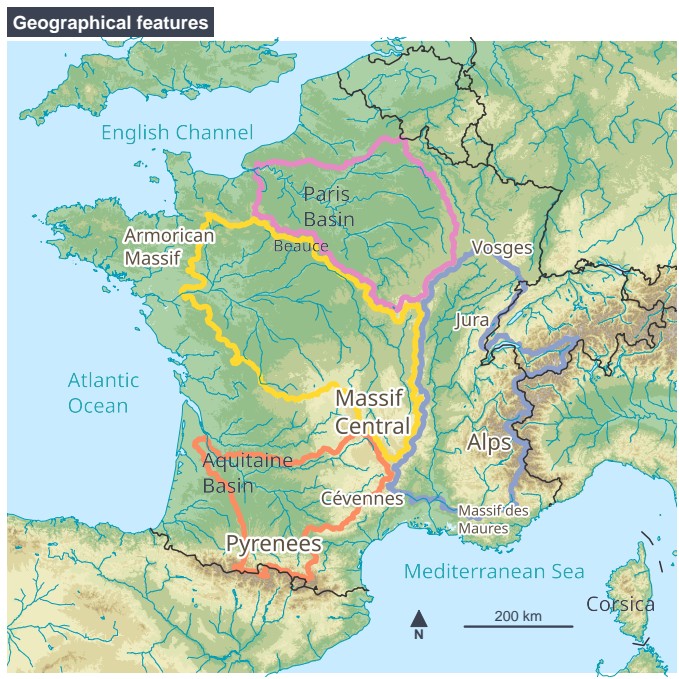

**Figure 1.** Map of the main geographical features of France: mountain ranges (text with white outline), regions (text without outline), and main basins (pink: Seine; yellow: Loire; orange: Garonne; purple: Rhône) (river network: Lehner and Grill, 2013; DTM: Lehner et al., 2008; shoreline & political boundaries: Wessel and Smith, 1996; NOAA, 2017).

## 2 French physical and hydroclimatic context

### 2.1 Physical characteristics

Metropolitan France (mainland France and Corsica; see Fig. 1), covers an area of 550,000 km² (expanding 1,000 km from north to south, and east to west), including Corsica, the fourth largest Mediterranean island (8,700 km²). It is mainly bounded by coastlines (with the North Sea, the English Channel, the Atlantic Ocean and the Mediterranean Sea) and shares terrestrial borders with eight neighboring countries (Belgium, Luxembourg, Germany, Switzerland, Italy, Monaco, Spain, Andorra). Metropolitan France offers a wide variety of natural landscapes inherited from several geological phases, giving rise to ancient (e.g. Armorican Massif, Massif des Vosges, and Massif Central) and younger mountain ranges such as the Jura, Alps and Pyrenees (see Fig. 2a). The average altitude is 344 m, and the elevation reaches a maximum of 4,806 m at Mont Blanc (in the Alps). These mountain ranges form the boundaries of several sedimentary basins, including the Aquitaine Basin in the southwest and the Paris Basin to the north.

The French geological diversity is illustrated with the wide range of colors found on the lithological map of France (see Fig. 2c). We give here a short description of the (complex) geology of France, focusing on its implications for understanding the hydrogeology of French catchments. A more detailed account is provided by Pelletier (2021):



– A quarter of France is covered by basement rocks that are metamorphic or igneous in origin. Basement formations host small local aquifers, but no large-scale aquifers. The two main areas are the Armorican Massif in the northwest, almost exclusively composed of ancient formations, and the Massif Central in the center of the country (a more complex area where Hercynian basement, sedimentary formations and recent volcanic formations coexist). Basement rocks also outcrop in smaller massifs of lesser extent (e.g. the Massif du Morvan in the upper Yonne basin, the Massif des Albères at
the border of France and Spain in Catalogne, the Massif ardennais around the Meuse River at the border with Belgium, the Massif de la Serre in the Jura Mountains near the Swiss border and the Massif des Maures on the Mediterranean coast near Saint-Tropez). Note that the Massif des Vosges is divided between basement formations to the south and sedimentary formations to the north;

     – Intensely folded areas are hydrogeologically very complex, because in these regions, aquifers, if they exist, are of limited
75       spatial extent and therefore difficult to identify for mapping on a regional or national scale. These areas correspond to the most recent massifs: the central and inner part of the Alpine arc, the Pyrenees and the Languedoc massifs;

     – The rest of the territory is occupied by two main types of formations: sedimentary formations, and alluvial plains. The sedimentary formations form two major basins: the Paris Basin, which covers a third of Metropolitan France, and the Aquitaine basin in the southwest. Formed by sedimentary deposits left by marine intrusions, they appear as a succession
80       of geological layers with diverse aquifer properties, whose outcrop areas form rings, known as the "pile d'assiettes" (plate stack) in the Paris Basin;

     – Alluvial plains and the accompanying aquifers are formed by the scouring and degradation by rivers of various materials, which are then deposited along the river bed. In mainland France, they are generally of limited geographical extent, unlike in other parts of the world. Two alluvial aquifers, however, are of greater importance: they are located in the upper Rhine
85       plain in Alsace, and in Bresse (around the Saône River, north of Lyon).

     – the variability of surface aquifer properties is clearly apparent when one looks at the density of the observed river network in Fig. 2c, where for example the large areas underlaid by chalk formation in the Paris Basin stand out with their much lower drainage density, while the Champagne Humide region, characterized by clayey and loamy soils, has a much denser river network. The Beauce region, underlaid by the Beauce limestones aquifer complex, is clearly visible as a
90       white spot southwest of Paris at the border of the Seine and the Loire basins.

   In 2015, farmland covered around 51 % of Metropolitan France, compared with around 40 % for soils not directly subject to anthropogenic pressures (woodlands, wetlands and water surfaces) and around 9 % for artificialized soils (see Fig. 2d).

## 2.2 Hydro-climatic characteristics

According to the Köppen-Geiger climate classification (see e.g. Peel et al., 2007; and Strohmenger et al., 2023 for a recent
analysis on France), more than 90 % of the French territory belongs to the *Cfb* class (i.e. temperate without dry season and with warm summers), with Corsica and the Mediterranean shore belonging to the *Csa* or *Csb* classes (temperate with dry





and respectively hot or warm summers). The mountain ranges belong to class *Dfb* (continental without dry season and warm summers), *Dfc* (continental without dry season and cold summers), and the high-elevation summits belong to the *ET* class (polar climate). Average annual precipitation ranges from 500 mm yr$^{-1}$ for the driest regions (e.g. Mediterranean coasts) to

more than 2,000 mm yr$^{-1}$ in mountainous regions (see Fig. 2e and 2f). While precipitation usually varies in a progressive manner over much of the country, a few mountain ranges are characterized by strong differences: one can mention for example the Cévennes range in the south, where average rainfall reaches 2,000 mm yr$^{-1}$, located not far from the Crau lowlands, where average rainfall is less than 500 mm yr$^{-1}$. A similar situation exists near the Rhine valley in Alsace, where average rainfall in the Massif des Vosges reaches 2,000 mm yr$^{-1}$ while it is less than 500 mm yr$^{-1}$ in the Colmar plain. As far as temperature

is concerned, it is probably enough to mention the mild winter temperatures on the ocean coast and the Mediterranean sea, the lowest temperatures being reached on the high mountain ranges where a few glaciers still exist despite the global warming trend.

The Metropolitan France's hydrographic network (Fig. 2b) is mainly organized around four major rivers (the Loire, Seine, Garonne and Rhône), whose catchments cover more than 60 % of mainland France. The Rhône and the Garonne catchments

are transboundary, with their sources in Switzerland and Spain respectively. Other transboundary rivers are found in France such as the Meuse (crossing France, Belgium and the Netherlands), the Rhine (crossing Switzerland, Liechtenstein, Austria, Germany, France and the Netherlands), and the Roya (crossing France and Italy), among others (e.g. the Sarre, the Oise, etc.). The French rivers are characterized by different hydrological regimes, with rain-dominated catchments, both rain- and snow-dominated catchments, snow-dominated catchments, Mediterranean catchments, and groundwater-dominated catchments. The

most upstream parts of a few mountainous catchments are still influenced by glaciers. In Metropolitan France, glaciers are located in the Alps and in the Pyrenees. Most of them cover rather small areas, especially when related to the area of catchments. The two main glaciers are the Argentière glacier and the Mer de Glace, in the Mont Blanc massif. Due to climate change, Vincent et al. (2019) showed that these two glaciers lost 34 and 45 m of water-equivalent depth, respectively, since the beginning of 20[th] century, representing 25 and 32 % of their thickness, respectively. Some other glaciers are located in the Ecrins and

Vanoise massifs.

Most of France's catchments are impacted by human activities through the presence of large dams (for water supply, hydropower or flood and low-flow management), river abstraction and groundwater pumping for agricultural, industrial, and drinking water use, therefore we aimed to exclude the most influenced catchments. The location of the main French reservoirs are shown in Fig. 2a. Note that a large number of small artificial water bodies exists in France, for various uses (recreation,

irrigation, boating, fishing, etc.), but often with more limited information available.

## 2.3 Main data producers

The French hydrosystems have been monitored for several decades through various observational networks, maintained by different organizations. These mainly include national agencies such as Météo-France for climatic data, the French geological survey (*Bureau de recherches géologiques et minières*, BRGM) for geological and hydrogeological information, the National

Institute of Geographic and Forest Information (*Institut national de l'information géographique et forestière*, IGN) for ge-

ographic and forest information, and different State services and independent producers (e.g. EDF, CNR, universities, etc.) for hydrological data. These streamflow data are made available by the Central service for hydrometeorology and inundation forecasting (*Service central d'hydrométéorologie et d'appui à la prévision des inondations*, SCHAPI).

## 3 Catchment boundaries

The first step to build the `CAMELS-FR dataset` was to delineate the contours of the catchments boundaries, which were then used to calculate hydroclimatic time series at this scale.

### 3.1 Making a flow direction grid to delineate catchment boundaries at national scale

Hydrological analysis and modelling require to associate climatic forcing averaged at the catchment scale with the streamflow measurements recorded at the outlet of a catchment. It was therefore necessary to identify the geographic extent of all French

catchments.

To do so, we used a flow direction grid, i.e. a matrix summarizing the topological relationships for Metropolitan France. This grid is derived from a topographic analysis of two digital terrain models (DTMs). For continental Metropolitan France, we used the DTM from the Shuttle Radar Topography Mission (SRTM) project with resolution of $100 \times 100$ m$^2$ (Rabus et al., 2003; Farr et al., 2007), and for Corsica, we used the BD ALTI v1.0 from the IGN (2001) with resolution of $25 \times 25$ m$^2$.

Because no DTM is error-free, we used an "observed" river network to force the geometry of the theoretical river network to be closer to reality. To constrain the flow directions via the stream-burning method, we used the CARTHAGE river network (French water agencies, 2017b) with removed channels. This method consists in initially burning the vector network in the DTM by artificially lowering the altitude of the pixels belonging to the network. For more details about the method, see Bourgin et al. (2010).

### 3.2 Geographic repositioning of hydrometric stations on the flow direction grid

When the metadata linked to the gauging stations are well filled in, the geographical coordinates (latitude, longitude and altitude) and the area of the upstream catchment of the station are available. However, this information is not always consistent with the flow direction grid that we made. Sometimes the resolution of DTM is not sufficient, sometimes the gauging station's coordinates are erroneous or not sufficiently precise. Repositioning geographically the hydrometric stations on the calculated

flow direction grid is therefore often necessary. Each hydrometric station must be linked with the "river" pixel of the flow direction grid. Only then can the contour and area of the catchment be delineated.

For the gauging stations that are not positioned in a "river" pixel, we searched in the twenty-four surrounding pixels (i.e. radius of 200 m for the continental Metropolitan France) the theoretical position of the gauging station by iteratively calculating the catchment area located upstream of each surrounding pixel. The objective is then to minimize the difference between the

area indicated by the data producer (if available) and the area calculated on the flow direction grid. This method allows to automatically place a large part of the hydrometric stations on the flow direction grid.

**Figure 2.** Maps describing Metropolitan France territory. (a) Elevation, main rivers and main dam locations (triangles) (river network: Lehner and Grill, 2013; DTM: Lehner et al., 2008; shoreline & political boundaries: Wessel and Smith, 1996; NOAA, 2017); (b) Observed river network (CARTHAGE: French water agencies, 2017b); (c) Lithology (GLiM v2.0: Hartmann and Moosdorf, 2012; *su*: unconsolidated sediment, *ss*: siliciclastic sedimentary rocks, *py*: pyroclastics, *sc*: carbonate sedimentary rocks, *sm*: mixed sedimentary rocks, *va*: acid volcanics, *vi*: intermediate volcanics, *vb*: basic volcanics, *pa*: acid plutonics, *pi*: intermediate plutonics, *pb*: basic plutonics, *mt*: metamorphics, *wb*: water bodies, *ig*: ice and glaciers, *nd*: non-defined); (d) Land cover (CLC: French Ministry of the Environment, 1990); (e) Mean annual total precipitation over the 1991-2020 period (SAFRAN: Quintana-Segui et al., 2008; Vidal et al., 2010); (f) Mean air temperature over the 1991-2020 period (SAFRAN).



However, errors still exist on some stations. Therefore, all outlets were checked manually using a graphical user interface (GUI) we developed for this specific goal (Génot and Delaigue, 2018). This GUI (see screenshot on Fig. 3; not made available with the dataset) provides means to analyze the geographical location of the hydrometric station by visualizing maps (e.g. maps, or aerial photos), and the comparison between the observed and the theoretical river networks, but also by facilitating the comparison of catchment areas.

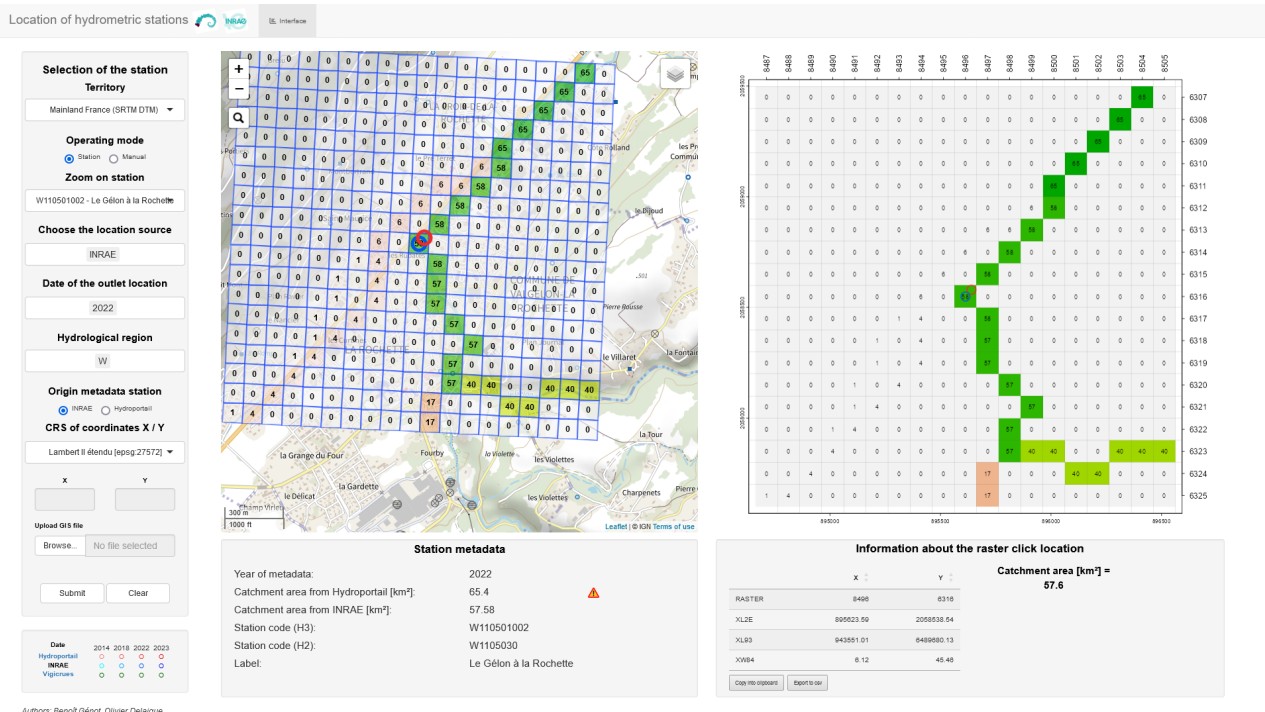

**Figure 3.** Screenshot of the graphical user interface helping to relocate hydrometric stations on the theoretical river network (Génot and Delaigue, 2018). The left panel is for option selection where we can select the territory (mainland France or Corsica), a specific station by providing its code or by directly inputing the coordinates, and the metadata year. The middle panel shows the zoomed location of the station on a map layer ("Plan IGN" v2 map layer, IGN, 2020), a superposition of the raster layer with contributing area grid to aid identifying the correct location of the hydrometric stations. The producer's gauging station is in red circle, and INRAE's location snaped on the theoretical river network is in blue circle. The metadata for the searched gauging station are displayed below this map. On the right panel the contributing area grid and the hydrometric station locations are displayed a new time allowing to click on the map, and to extract the raster information such as pixel location and its corresponding catchment area.




## 4 Selection of the catchment set

The second step to build the `CAMELS-FR dataset` was to select the catchments based on the following criteria: (i) hydrometric time series availability (Sect. 4.1), (ii) artificial reservoirs influences (Sect. 4.2), (iii) consistency in catchment areas 170 (Sect. 4.3), (iv) streamflow quality inspection (Sect. 4.4).

The application of the different criteria and analyses described in the following subsections resulted in the selection of 654 catchments, constituting the `CAMELS-FR dataset` v1.

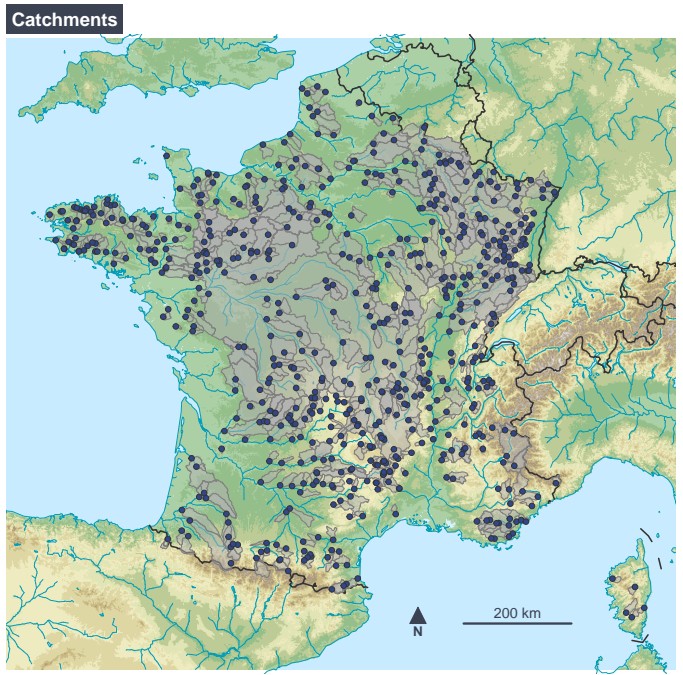

**Figure 4.** Location map of the 654 catchments and their outlets of the `CAMELS-FR dataset` v1 (river network: Lehner and Grill, 2013; DTM: Lehner et al., 2008; shoreline & political boundaries: Wessel and Smith, 1996; NOAA, 2017).

### 4.1 Selection on hydrometric time series availability

We excluded catchments with less than 30 years of complete data over the 1970-2021 period. This record length was arbitrarily 175 chosen because it allows robust statistical analyses. A year was considered complete when it had less than 20 % of missing data. This criteria was responsible for removing approximately 70 % of all stations available in France.

### 4.2 Artificial reservoirs influences

Our objective was to only select catchments displaying a level of human disturbance as low as possible. Here, we only consider influences due to artificial reservoirs. Therefore, the MADAM dataset v1.0 (Delaigue et al., 2024b) was used to estimate these



influences. To build this dataset, two sources of information were used, namely the work of Payan (2007) & Payan et al. (2008), and the GEOBS datasets (OFB and partners, 2023), in order to provide locations and volumes of dams in Metropolitan France (mainland and Corsica). Each time, we calculated a theoretical level of influence by estimating the equivalent water depth of the total water storage capacity of all dams on the catchment (sum of capacities divided by catchment area).

We cross-checked the two sources of information. The positions of the dams were manually checked. When the storage
capacity was inconsistent among the datasets, we validated the dam volume using the website of the French Committee for Dams and Reservoirs (*Comité français des barrages et réservoirs*, CFBR, 2023), by collecting information from various technical documents of the operators freely available on the Internet, or by directly contacting the operators.

Only the catchments with an equivalent water depth of storage capacity lower than 10 mm were selected, thus removing catchments considered highly influenced from our selection. This threshold comes from experience on past studies on this
issue (e.g. Payan et al., 2008), but remains arbitrary. The actual level of influence may depend on the local context (location of the dam(s) within the catchment, management objectives of the dam(s), etc.). But we preferred to stick to a single threshold value to have an homogeneous selection criterion.

Note that other types of influence may significantly modify natural streamflow in the selected catchments, typically water withdrawals for various uses or inter-basin water transfers. However since this information could not be accessed and processed
easily, it was not considered here as selection criterion. Thus, the CAMELS-FR catchment set should not be considered as strictly non-influenced catchments. For this reason, we also include in the catchment attributes descriptors giving the qualitative estimation of the local and general impact of human influences as provided by the data producers.

### 4.3 Consistency in catchment areas

The repositioning of the catchment outlets (reported by the data producers, alongside the catchment area) on the flow direction
raster was manually checked. We wished to have similar catchment areas between the automatically computed procedure using the flow direction raster and the information provided by the data producers. The catchments whose areas differ by more than 10 % were discarded from the selection. Further investigation would be needed to clarify the reasons for these differences, which may lead to have additional catchments in the CAMELS-FR dataset in the coming years. For example, there may exist two types of catchment area (topographic and hydrogeologic) for karstic catchments in the metadata provided by the
producers.

### 4.4 Streamflow quality inspection

Finally, a visual analysis of streamflow time series was performed in order to identify obvious errors in the streamflow series, such as flow interpolation, sudden drops and noises not referenced as so. The time series were evaluated by four observers. If at least two observers considered a time series to be incorrect, it was removed from the dataset. If only one observer deemed a
time series incorrect, it underwent group re-evaluation to get a consensus on the data quality. Catchments with such errors have been discarded from the dataset, pending for data correction from the data producers. This qualitative analysis of streamflow series may lead the addition (or removal) of catchments in the CAMELS-FR dataset in the coming years. The difficulty

of detecting non-natural records in streamflow time series has been recently illustrated and discussed by Strohmenger et al. (2023) on a large set of French catchments.

## 5 Hydroclimatic time series

### 5.1 Climatic time series

The SAFRAN atmospheric reanalysis (Quintana-Segui et al., 2008; Vidal et al., 2010) was used as source of daily climatic time series, aggregated at catchment scale. SAFRAN is a mesoscale analysis system of near-surface atmospheric variables from the SAFRAN-ISBA-MODCOU hydrometeorological model chain (SIM2; Le Moigne et al., 2020). The SAFRAN data are provided annually by Météo-France at the daily time step from 1958 to present, and there is no missing data. The reanalysis methodology is regularly updated and data is updated retroactively(since 1958). SAFRAN uses surface observations combined with data from meteorological models, in particular the ERA reanalysis of the European Centre for Medium-Range Weather Forecasts (ECMWF). Climate variables are analyzed by 300 m altitudinal steps. They are then interpolated on a regular grid with resolution of $8 \times 8$ km$^2$ (note that catchment rainfall is interpolated from ground measurements). The climatic data cover the whole metropolitan territory (mainland France and Corsica), as well as areas at the borders to correspond to the catchments of rivers flowing into France, with a total of 9,892 pixels. An evaluation of the SAFRAN reanalysis is provided in Vidal et al. (2010).

The SAFRAN variables used in the `CAMELS-FR dataset` are: solid precipitation, liquid precipitation, air temperatures (minimum and maximum), wind speed, specific air humidity, atmospheric and visible radiations. In addition, these climatic data feed a soil-vegetation-atmosphere (SVAT) model called ISBA (Le Moigne et al., 2020) to produce surface variables, such as snow water equivalent and a soil wetness index. Table 1 lists these variables and if they represent daily means or daily accumulations. Note that the aggregation time window may differ between the variables considered, which may have an impact on modelling results.

Using these variables, we calculated (i) three time series of potential evaporation estimates using three commonly-used formulas: Penman (Penman, 1948), Penman-Monteith (Monteith, 1965) and Oudin (Oudin et al., 2005), and (ii) a soil moisture index as computed by the GR4J rainfall-runoff model (Perrin et al., 2003, computed with a reservoir of 275 mm). Note that our versions of Penman and Penman-Monteith formulas use a snow-dependent albedo.

The catchment-scale climatic time series are obtained by spatial aggregation of the SAFRAN pixels intersecting the catchment contours. The SAFRAN variables are weighted-averaged using the pixel percentage within the catchment contour.

### 5.2 Hydrometric time series

The SCHAPI released in January 2022 the Hydroportail application for accessing hydrometric data at the national scale. Hydroportail classifies the hydrometric entities into three levels: hydrometric site, hydrometric station and hydrometric sensor. The hydrometric site refers to a portion of a watercourse on which flows are considered homogeneous. Several hydrometric





**Table 1.** List of daily SIM2 variables used in the `CAMELS-FR dataset`, and their aggregation methods at the daily time step.

| SIM2 product | Variable | Unit | Aggreg. methods | Aggreg. time window [UTC] |
|---|---|---|---|---|
| SAFRAN | Solid precipitation | $\mathrm{mm\,d^{-1}}$ | Accumulation | 06 h–06 h |
| SAFRAN | Liquid precipitation | $\mathrm{mm\,d^{-1}}$ | Accumulation | 06 h–06 h |
| SAFRAN | Air temperature | °C | Mean | 00 h–24 h |
| SAFRAN | Daily minimum air temperature | °C | - | 00 h–24 h |
| SAFRAN | Daily maximum air temperature | °C | - | 00 h–24 h |
| SAFRAN | Wind speed | $\mathrm{m\,s^{-1}}$ | Mean | 00 h–24 h |
| SAFRAN | Specific air humidity | $\mathrm{g\,kg^{-1}}$ | Mean | 00 h–24 h |
| SAFRAN | Atmospheric radiation | $\mathrm{J\,cm^{-2}\,d^{-1}}$ | Accumulation | 00 h–24 h |
| SAFRAN | Visible radiation | $\mathrm{J\,cm^{-2}\,d^{-1}}$ | Accumulation | 00 h–24 h |
| ISBA | Snow water equivalent | $\mathrm{mm\,d^{-1}}$ | Accumulation | 06 h–06 h |
| ISBA | Soil wetness index | – | Mean | 06 h–06 h |

(gauging) stations can be associated with a single site. A hydrometric station may include several sensors. The main benefit

of dividing the hydrometric entities into these levels is to give the producers the possibility to set a calendar (i.e. periods of validity of each station and sensor) at the site level and unify the corresponding streamflow data to produce a consolidated time series with a better qualification procedure.

The `CAMELS-FR dataset` daily streamflow time series were retrieved from the Hydroportail website using the hydroportail R-package (Delaigue, 2022). Streamflow data were retrieved at the station level to get all the data even for sites

where calendar information is missing or incomplete. For stations with sub-daily streamflow data, mean daily streamflows are calculated using a trapezoidal method, thus assuming a linear variation in streamflow between successive instantaneous streamflows. This was done directly by the data producer or by the Hydroportail procedure.

Streamflow data are available in the `CAMELS-FR dataset` in $\mathrm{l\,s^{-1}}$ and were also converted in water depth (in $\mathrm{mm\,d^{-1}}$) using the catchment area derived from the DTM (and not producer's area). As climatic data, the streamflow time series are

provided at the daily, monthly and yearly time steps.

### 5.3 Rules for time series aggregation

The time series are provided at the daily, monthly and yearly time steps. For the calculation of monthly time series, a month was considered missing if it had more than 3 days of missing data. The time series at yearly time step are provided using hydrological year (considered to start on 1 October) as basis of aggregation. Therefore the hydrological year of 1970 (first year

of the dataset), which starts on 1 October 1969, and is deemed incomplete because of the three first missing months. Moreover, these yearly data are computed based on the above-mentioned monthly times series. If one month is missing, the whole year is considered missing.





**Table 2.** Number of catchment attributes in each class for the `CAMELS-FR dataset` v1.

| Class | Nb. |
| --- | --- |
| Location & Topography | 91 |
| Climatic indices | 51 |
| Hydrometry | 18 |
| Hydrological signatures | 24 |
| Hydrogeology | 11 |
| Geology | 17 |
| Soil | 25 |
| Land cover | 10 |
| Intervention degree | 7 |
| Other | 1 |

## 6 Catchment attributes

The `CAMELS-FR dataset` v1 contains 255 attributes, organized in 10 classes (see Tab. 2), described in the following sub-
sections. A spreadsheet listing each catchment attribute is included in the `CAMELS-FR dataset`.

### 6.1 Location

Most of the location attributes are information given by the data producer and extracted from Hydroportail. Such attributes
include the station codes, names, location, type of station, etc. The data producer gives, for most catchments, an estimate of
catchment area. Note that for some catchments this area is significantly different from the one derived from the DTM analysis
(see Sect. 4.3). Other catchment attributes were computed for the `CAMELS-FR dataset`. They include station coordinates
(after repositioning on the DTM-derived river network), catchment area (derived from the DTM), and information on the station
nestedness (indicating whether the catchment is nested, the number of stations downstream, etc.).

### 6.2 Topography

Two DTMs (NASA & NGA DTM from SRTM for catchments located on the continent and the BD ALTI v1.0 (IGN) for
catchments in Corsica) were used to estimate various attributes describing catchment topography, drainage density, and mor-
phometry. Among the topography attributes are the mean and the percentiles distribution for elevation, slope, distance to
catchment outlet and topographic index. The methodology used to calculate the topographic index follows Ducharne (2009),
which reformulates the index used in TOPMODEL (Beven and Kirby, 1979) to become a dimensionless index. The percentage
of catchment slope classes (flat, gentle, moderate, strong, steep, and very steep) and orientation, and the classical drainage
density which is measured as the ratio of total length of stream channels to catchment area (Horton, 1932) are also provided.
Among the morphometry attributes, several catchment shapes indicators are provided. These include three variants of basin



form factor (Horton, 1932; Zăvoianu, 1978, cited by Zăvoianu, 1985, p. 109; Subramanya, 2013, p. 172), the compactness coefficient (Fitzpatrick, 2017), the circularity ratio proposed by Miller (1953, cited by Zăvoianu, 1985, p. 104), the catchment relief ratio (Fryirs and Brierley, 2013, p. 35), and two variations of elongation ratio: (i) using a circle as a reference (Schumm, 1956), and (ii) using the catchment area as a reference (Fryirs and Brierley, 2013, p. 34).

### 6.3 Climatic indices

Numerous climatic attributes were estimated: annual mean of different variables (air temperature, potential evaporation, precipitation) over the entire studied period, aridity and seasonality indices, measure of the asynchronicity between the precipitation and potential evaporation, etc. The attributes that use potential evaporation are computed three times using Penman, Penman-Monteith and Oudin formulas. Attributes on heavy-precipitation days frequency (e.g. frequency of high-precipitation days, i.e. larger than 5 times the mean daily precipitation) and intensity (strongest rainfall on record), and dry-days frequency (average duration of dry periods, i.e. number of consecutive days with precipitation lower than $1\ \mathrm{mm\,d^{-1}}$) were also computed. Time series of statistics at annual time step using hydrological year are also provided including annual daily maximum precipitation and annual seasonality index. As mentioned in Sect. 5.3, the first year (1970) is always missing. Furthermore, maps with resolution of $1 \times 1\ \mathrm{km^2}$ of hourly and daily rainfall intensity, developed by Arnaud et al. (2008) from the SHYREG-Pluie database (Base de données SHYREG-Pluie © INRAE, 2016, all rights reserved), were used to calculate a normalized indicator of rainfall intensity at the catchment scale. This indicator is defined as the ratio of the hourly to daily rainfall intensity with 10-year return period (Poncelet, 2016).

### 6.4 Hydrometry

Several `CAMELS-FR dataset` attributes are related to streamflow data quality estimated for each gauging station. Some of these attributes are provided by the data producers. For example, the overall level of uncertainty is qualified by the producer for three flow ranges (high, mean and low flow). Moreover, several quality codes are associated to each streamflow value, depending on the way the value was estimated (classical flow estimation using a water level measurement and a rating curve, or a value reconstituted a posteriori) (see Sect. 5.2).

Note that this information lacks consistency throughout the `CAMELS-FR` catchment set, since this qualification is done by regional services, which have different qualification methods according to the regional hydroclimatic context.

Additional attributes were estimated for each station to describe the number of gaps over the studied period (e.g. percentage of missing data in the total period (1 January 1970 to 31 December 2021) and the overall percentage of streamflow data flagged as questionable or as unqualified by the data producer. A simple algorithm was also applied to identify potential errors due to linear interpolation between two values.

Finally, we attributed to each station an overall estimation of low-flow quality was made based on the visual analysis of temporal streamflow series: stations with low flows showing suspicious data or behaviour have been identified, in the `CAMELS-FR dataset` v1 showing 52 % of catchments with such issues.



## 6.5 Hydrological signatures

As for catchment climate, various hydrological attributes were estimated to describe the main catchment hydrological features. These include catchment aridity (i.e. the ratio of mean daily potential evaporation to mean daily precipitation), catchment yield, streamflow elasticity to precipitation, catchment seasonality (e.g. month with the minimum mean monthly streamflow). The baseflow index (BFI) was estimated following three different approaches: (i) method proposed by Ladson et al. (2013) that uses a digital filter, (ii) method proposed by Gustard and Tallaksen (2008) through linear interpolation of a five-day non-overlapping

streamflow minima (computed with the lfstat R-package; Laaha and Koffler, 2022), and (iii) method proposed by Pelletier and Andréassian (2020) that uses a conceptual quadratic reservoir model (computed with the baseflow R-package; Pelletier et al., 2021). For the latter, before calculating the BFI, we filled the gaps in time-series with GR6J-CemaNeige (Pushpalatha et al., 2011; Valéry et al., 2014a, b), using Penman (Penman, 1948), Penman-Monteith (Monteith, 1965), and Oudin (Oudin et al., 2005) formulas. However, we provide the BFI using only the Oudin potential evaporation formula because there were no

significant BFI differences when using other formulas. The other attributes that use potential evaporation are computed three times using the three selected formulas. Time series of statistics at annual time step are also provided for annual maximum daily streamflow and the mean monthly annual minimum streamflow (QMNA) using hydrological and calendar year respectively.

## 6.6 Hydrogeology

The average catchment permeability and porosity were estimated thanks to the GLobal HYdrogeology MaPS database (GL-
HYMPS v2.0) (Huscroft et al., 2018). The national hydrogeological reference map (BDLISA v3) (Brugeron et al., 2018; BRGM, 2022) was used to calculate the karstic portion of the catchments and the percentages of each catchment covered by different hydrogeological formations (e.g. bedrock, sedimentary, alluvial zones).

## 6.7 Geology

The Global Lithological Map database (GLiM v1.0) (Hartmann and Moosdorf, 2012) was used to characterize the catchment
geology. This database, available at the 0.5° spatial resolution, provided estimates of (i) the dominant geological class of each catchment and (ii) the percentages of different lithologies (e.g. metamorphics, carbonate sedimentary rocks, etc.; see Fig. 2c).

## 6.8 Soil

Soil characteristics were described using four datasets. First, the Global 1-km Gridded Thickness of Soil, Regolith, and Sedimentary Deposit Layers v1 produced by Pelletier et al. (2016) was cropped in order to estimate, for each catchment, (i) the
mean soil depth and (ii) the distribution (percentiles) of the soil depths over the catchment. The European Soil Database Derived data (ESDD) (JRC et al., 2013a, b) provides soil attributes with resolution of $1 \times 1$ km$^2$ for a topsoil layer and a subsoil layer with the boundary at 30 cm soil depth for clay, sand, silt and organic carbon content, bulk density, coarse fragments and total available water content (TAWC). We therefore, aggregated these two layers from ESDD using weighted arithmetic mean in cells based on their soil depth for each pixel. We consider the attribute "depth available to roots" as the correspondent





soil depth for each pixel. For the aggregation of TAWC, instead of calculating the weighted arithmetic mean, we calculated the sum of the two layers. Then we cropped the raster for each catchment to extract the percentiles, mean and skeweness of each attribute for the topsoil and the aggregated layer. The EU-SoilHydroGrids v1.0 (Tóth et al., 2017) with resolution of $250 \times 250$ m$^2$ was used to extract the saturated hydraulic conductivity. This dataset has seven soil layers at at 0, 5, 15, 30, 60, 100, and 200 cm depth. The first four layers, that are less than or equal to 30 cm depth, are considered as topsoil. Therefore,

we aggregated these layers to provide information on topsoil (using the first four layers) and the total seven layers as well using depth-weighted harmonic mean. The hydraulic saturated conductivity was divided by 24 in order to provide the results in cm/hour, and then was cropped for each catchment to extract the percentiles, mean and skeweness. Finally, the TAWC was also estimated by INRAE over France from the Réservoir utile des sols de la France métropolitaine v1.2 database (Roman Dobarco et al., 2021; Le Bas, 2021) with resolution of $90 \times 90$ m$^2$, providing another distribution of TAWC at the catchment scale.

### 6.9 Land cover

The CORINE Land Cover dataset was used to estimate, for each catchment, (i) the dominant land cover class and the (ii) percentage of each class, at two dates (French Ministry of the Environment, 1990; French Ministry of Ecology, 2018). This analysis has been performed for all aggregation levels (1, 2 and 3) of the CORINE Land Cover classification.

### 6.10 Intervention degree

Influence level attributes gather (i) information given by the data producer on the estimated anthropogenic impacts on the observed streamflow time series (4 attributes) and (ii) metrics we calculated to estimate the influence of dams present within each catchment (3 attributes). Thus, we listed, for each catchment, the number of dams and the total volume of artificial storage.

## 7 Dataset description and add-on products

### 7.1 Illustration of several attributes of the CAMELS-FR dataset

Figure 5b shows the distribution of four climatic characteristics (mean annual precipitation, mean annual potential evaporation, seasonality index and max annual precipitation) of the `CAMELS-FR dataset` catchment set, grouped by hydrological region (see Fig. 5a). The northwestern catchments have lower annual precipitation than those in the south, the latter being characterized by high variability in annual precipitation and also in mean annual potential evaporation. Southern catchments show lower seasonality index values, due to Mediterranean climate characterized by winter maximum for precipitation and sig-

nificant summer drought (de Lavenne and Andréassian, 2018). Figure 5c shows each `CAMELS-FR dataset` catchment in a Budyko-type nondimensional space (Andréassian and Perrin, 2012) and for each hydrological region, showing the diversity of hydroclimatic context. Several catchments are located over the water limit ($Q = P$), or under the energy limit ($Q = P - PET$) due to potential missestimation of precipitation, potential evaporation or streamflow, errors in catchment area estimation, or non-conservative catchment behavior.



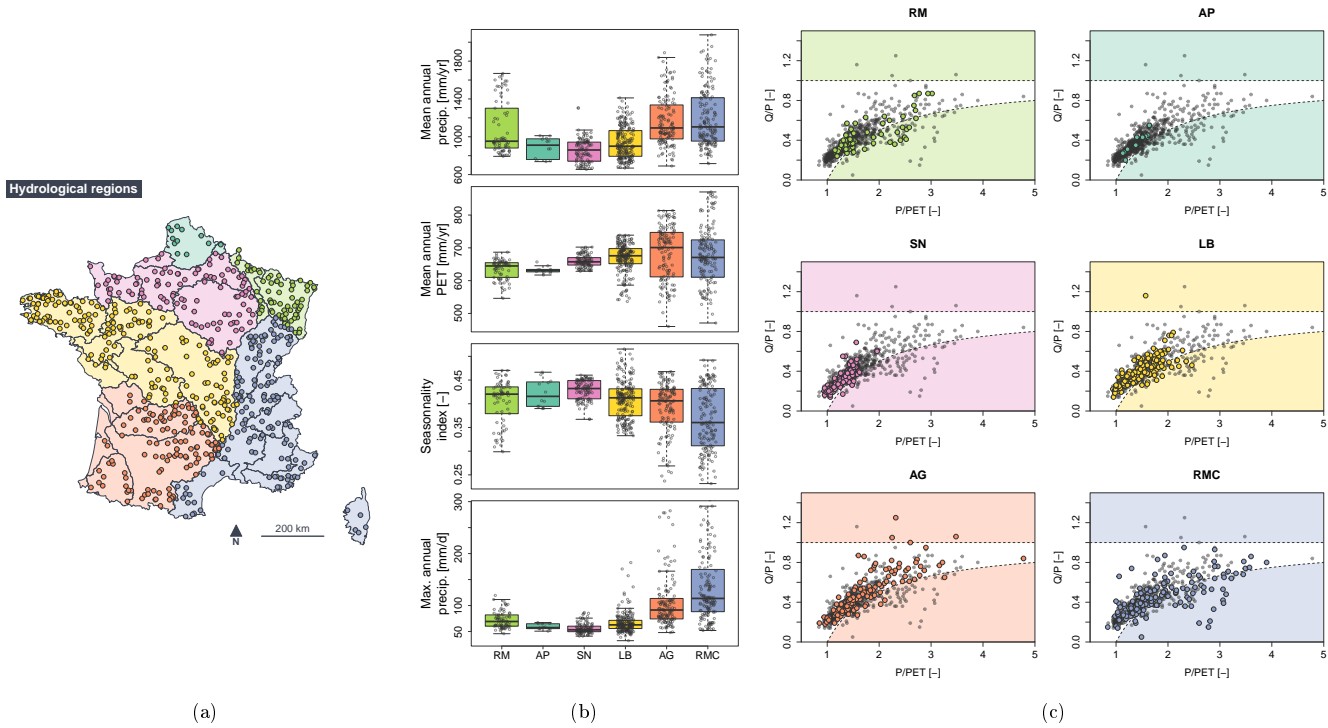

(a)                          (b)                          (c)

**Figure 5.** Distribution of `CAMELS-FR` catchment set over hydrological regions: (a) Geographical locations; (b) Boxplots of climatic indices; (c) Projection on a Budyko space (AG: Adour-Garonne, AP: Artois-Picardie, LB: Loire-Bretagne, RM: Rhin-Meuse, RMC: Rhône-Méditerrannée-Corse, SN: Seine-Normandie; hydrological region boundaries: French water agencies, 2017a).

Figure 6 presents maps and distributions of three climatic attributes (mean annual precipitation, mean annual potential evaporation and seasonality index: one can clearly identify the regions with higher than average mean precipitation (Fig. 6a): Brittany in the West, and all the mountain ranges, i.e. Vosges, Jura, Alps, Massif central and the Pyrenees. Note that the spatial distribution of extreme precipitations shows a clear structure along a northwest-southeast axis (Fig. 6b), while potential evaporation is structured along a north-south axis, with some variations induced by the mountain ranges (Fig. 6c). Streamflow-

related indices (Fig. 6d, 6e, 6f) do not show clear large-scale patterns, because they reflect the complex interplay between the climatic factors and the pedo-lithologic factors (see Fig. 2b & 2c).

**7.2    CAMELS-FR add-on products**

Two side-products were developed around the `CAMELS-FR dataset`, to provide tools to visualize catchment-scale time series, and synthetic summaries, to give the user a good overview of the data.

–  `CAMELS-FR graphical fact sheets` (Delaigue et al., 2024a): static plots summarizing hydro-climatic, topographical, hydrogeological, and land cover data (available in English and French) (see example Fig. 7);

**Figure 6.** Maps and distributions of few catchment-scale attributes (hydrological region boundaries: French water agencies, 2017a). (a) Mean annual precipitation; (b) Mean annual daily maximum precipitation ; (c) Mean annual long-term potential evaporation (Oudin method); (d) Mean annual streamflow; (e) Mean annual minimum monthly streamflow (QMNA); (f) Baseflow index using (Pelletier and Andréassian, 2020); (g) Seasonality index (de Lavenne and Andréassian, 2018); (h) Percentage of catchment that is likely karstic; (i) Mean elevation of the catchment. Note that for those indices requiring a potential evaporation estimate, we used consistently the Oudin method.

– `CAMELS-FR time series dynamic graphs` (Delaigue et al., 2024d): HTML pages with dynamic plots of hydroclimatic time series (click and drag zooming, and "onmouseover" display of legends and values; available in English and French) (see example Fig. 8).

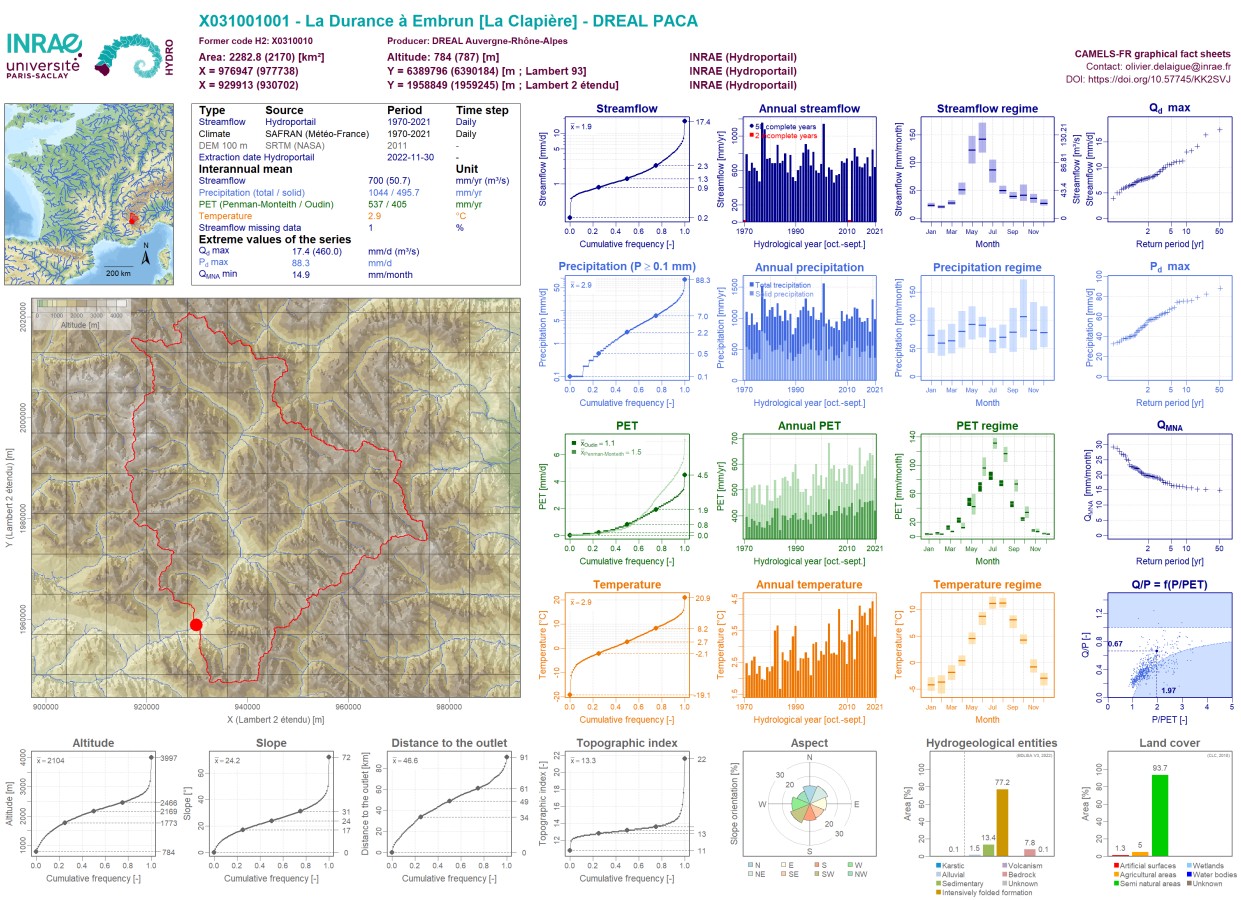

**Figure 7.** Example of `CAMELS-FR graphical fact sheets` (Delaigue et al., 2024a) for the X031001001 station ("La Durance à Embrun [La Clapière] - DREAL PACA").

## 8 Conclusions and perspectives

The `CAMELS-FR dataset` v1 gathers data over 654 catchments located in France, including daily hydroclimatic time series over the 1970-2021 period and 255 attributes split in 10 classes (see Tab. 2). These catchments represent a significant diversity of hydroclimatic contexts (e.g. snow- or groundwater-dominated catchments, Mediterranean catchments, etc.). The catchment selection was based on four criteria: (i) streamflow data availability over the 1970-2021 period, (ii) limited artificial influences (quantified based on upstream dam storage only), (iii) consistency between catchment area provided by the data producer





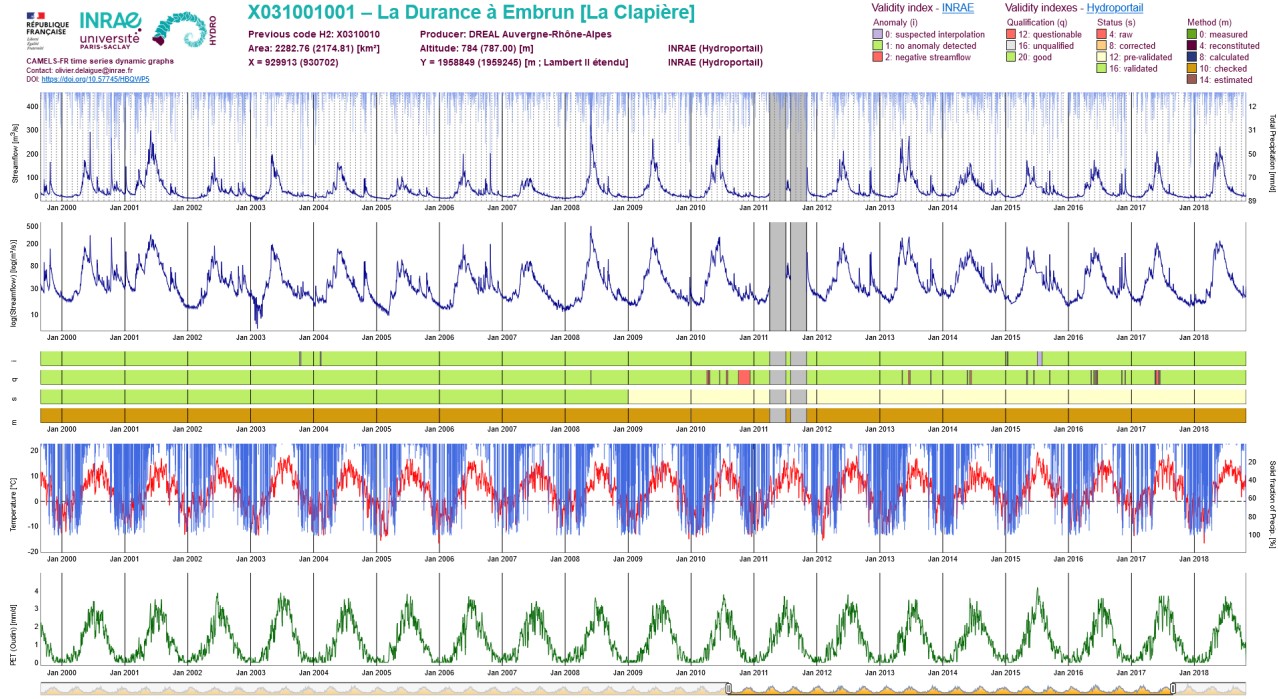

**Figure 8.** Example of `CAMELS-FR time series dynamic graphs` (Delaigue et al., 2024d) for the X031001001 station ("La Durance à Embrun [La Clapière] - DREAL PACA").

and estimated by the DTM analysis and (iv) visual analysis of the streamflow time series (at the daily time step). We did not consider any selection criteria based on hydrological model efficiency.

The `CAMELS-FR dataset` has been designed to be a "living" dataset. Several changes and updates are planned in subsequent versions: time series lengthening, streamflow values correction by the data producers, addition of "new" catchments

(e.g. from French overseas territories). Versions with data at finer temporal and spatial resolutions may also be developed. A web-app devoted to the selection of catchment subsets using hydro-climatic, topographical, hydrogeological, or land cover criteria is currently under development. A `CAMELS-FR dataset` extension into the world-wide Caravan initiative (Kratzert et al., 2023) is also foreseen.

## 9 Code availability

The code sources are detailed in the description file available in the data repository (see below).

## 10   Data availability

The `CAMELS-FR dataset` (Delaigue et al., 2024c) can be freely downloaded from the French governmental research data warehouse entrepot.recherche.data.gouv.fr using the following doi: 10.57745/WH7FJR. The `CAMELS-FR time series dynamic graphs` (Delaigue et al., 2024d) and the `CAMELS-FR graphical fact sheets` (Delaigue et al., 2024a)
also have their own digital object identifier, respectively doi: 10.57745/KK2SVJ and doi: 10.57745/HBQWP5.

Most of the variables included in the dataset are based on the French State Open License version 2.0 (https://www.etalab. gouv.fr/licence-ouverte-open-licence, last access: 23 October 2024), which is compatible with the CC BY license. A few are based either on CC BY licenses version 3 or 4, or on specific licenses (provided with the dataset). The `CAMELS-FR dataset` is distributed under the CC BY license version 4.0. (https://creativecommons.org/licenses/by/4.0, last access: 23 October 2024).
Each modification to the dataset can be tracked, as the version number will be automatically updated. Older releases will remain available, even if the data has been updated. A "NEWS" file will be updated in order to track changes between versions.

*Author contributions.*   OD conceptualized the work. OD, GG, BG, PB, VA, and NA wrote the computer codes to format the data and calculate the indicators. OD, GG, PB, and CP visually inspected the streamflow time series. OD, GG, and PB assessed the locations and volumes of French reservoirs. OD, GG, PB, CP, and VA drafted the manuscript. All authors reviewed and edited the manuscript.

*Competing interests.*   The contact author has declared that none of the authors has any competing interests.

*Acknowledgements.*   The authors would like to thank colleagues from SCHAPI and DRIEAT: Jean-Nicolas Audouy, Carine Chaléon, and Stéphanie Pitsch for their essential help on the Hydroportail data. We would like to thank hydrological data producers for responding to our numerous requests to check, edit, and if necessary, correct Hydroportail data. We would like to thank Météo-France: Pierre Etchevers for supporting this endeavour, François Besson and Jean-Marie Willemet for providing data. We would like to thank OFB: Karl Kreutzenberger
for providing the GEOBS data, and Pierre Steinbach for his expertise on this database and for taking into account our suggestions of modifications. We would like to thank many current or former members of the HYCAR research unit at INRAE for their contributions to this database: François Bourgin, Pierre-Yves Bourgin, Mathilde Chauveau, Louise Crochemore, Andrea Ficchì, Carina Furusho-Percot, Nicolas Le Moine, Laure Lebecherel, Florent Lobligeois, Pierre Malassenne, Pierre Nicolle, Julien Peschard, Carine Poncelet, Maria-Helena Ramos, Gaëlle Tallec, Guillaume Thirel, and Julie Viatgé. We wish to thank Jean-Nicolas Audouy, François Bourgin, Stéphanie Pitsch, and
Guillaume Thirel for their comments on the draft version of this manuscript.



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
