# Peer review of "CAMELS-FR dataset: A large-sample hydroclimatic dataset for France to explore hydrological diversity and support model benchmarking"

_Earth System Science Data, 2024_

## Author Comment (AC1)

**RC1 answer**

The authors thank Prof. Hrachowitz for his kind and encouraging words on this work.

**(1) It would be helpful for the reader to add a very brief description of how CAMELS-FR complements the recently published EStreams data set (Do Nascimento et al., 2024)**

We indeed see CAMELS-FR and EStreams as complementary datasets. Their main differences lie in the scale at which they were designed (national/continental), which led to different catchment selection methods, data analyses and choices in data sources. The main difference are:
- Data quality checks: our approach consisted in a reproducible selection of catchments based on a few objective criteria, where, at some point of the selection process, the human eye was involved. We even organized our perusal to have several independent human eyes doing the same analysis. While this is obviously not sufficient to exclude all the stations with data-quality issue, it is in our opinion an irreplaceable procedure before distributing a dataset for research purposes. For example, we decided to provide time series and not only give a tool to download the data, because as the producers update their data, the streamflow can change from one extraction to another from the national streamflow archive. They regularly check and fix some time steps in the series, or update the rating curves. This could impact the reproducibility of the hydrological signatures computation for one catchment from one extraction to another. In addition, we visually inspected the time series of all the catchments while taking into account the metadata and the producer comments to exclude influenced catchments (we know by experience that the producer flags are not sufficient to identify the influenced catchments, as sometimes they only mention the influence in the comments of the station).
- Catchment selection: Most of the 654 stations of CAMELS-FR are indeed present in the EStreams dataset. Our objective in CAMELS-FR was to include only stations having hydrological significance (e.g. excluding canals) and with limited known artificial influences (e.g. not closely downstream of dams). The detailed analyses and exchanges with data producers needed to conduct this detailed catchment selection is probably not feasible at the scale of a European dataset like EStreams.
- Climate data sources: In CAMELS-FR, we preferred to use national climatic products provided by experts from Météo-France as we feel this is the highest quality-level we can achieve currently. Their resolution (8 km) is better than the E-OBS dataset whose 25-km resolution could be problematic for small catchments in mountainous regions. It is tempting to use a single product for Europe, but since the density of raingauges changes so much between countries, it does not warrant any homogeneity in catchment precipitation estimation.
- Catchment descriptors: We acknowledge that the EStreams dataset provides some additional indicators that are not included in CAMELS-FR dataset such as NDVI, LAI, snowcover, etc. However our dataset provides more metadata, and more detailed metrics about topography, climate, hydrology, hydrogeology and soil. So the two datasets are probably complementary on this aspect.

We will mention the EStreams dataset in the introduction of the revised article and shortly explain the main differences with CAMELS-FR (end of line 42 of the paper).
"As a national dataset, CAMELS-FR should also be seen as complementary to other datasets built at larger scales that include France, e.g. the EStreams dataset at European scale (Do Nascimento et al., 2024). CAMELS-FR differs from such datasets in the criteria used in the catchment selection process, the data analysis methods and the choices in data sources that may be available at national but not larger scales."

**(2) not sure what the different symbol sizes in Figure 5 indicate. Catchment area? Please add this information in the Figure caption**

The small dark grey points represent all the catchments outside the given hydrological region. The bigger colored ones represent the catchment in the given hydrological region. We will add the definition in the caption:
"Figure 5. Distribution of CAMELS-FR catchment set over hydrological regions (AG: Adour-Garonne, AP: Artois-Picardie, LB: Loire-Bretagne, RM: Rhin-Meuse, RMC: Rhône-Méditerrannée-Corse, SN: Seine-Normandie; hydrological region boundaries: French water agencies, 2017a): (a) Geographical locations; (b) Boxplots of climatic indices; (c) Projection on a Budyko space (large colored dots: catchments inside the given hydrological region; small grey dots: all the other catchments)"

---

## Author Comment (AC2)

**CC1 answer**

We thank Joseph Janssen for his feedback on the paper.

**In section 4, can you specify how many catchments failed each of your four catchment selection criteria (i.e., area mismatch). Thanks!**

As we envision this dataset to be updated regularly, we did not want to be too specific to a single version of the dataset regarding the exact number of catchments remaining for each selection criteria. However, we include in the paper the criteria that had the most impact on the station selection where we mention that the time series availability was responsible for removing approximately 70 % of all stations available in France (see Sect. 4.1). To appease your curiosity, we detail below how many catchments passed each of the four catchment selection criteria. When we made the data extraction, there were 4667 hydrometric stations.

| Compliance to criterion | Hydrometric time series availability | Artificial reservoirs influences | Consistency in catchment areas |
|---|---|---|---|
| True | 1313 | 3669 | 4572 |
| False | 3354 | 998 | 95 |

Following the order presented in the paper, the number of stations at each selection criterion step is:

1. Hydrometric time series availability: 1313
2. Artificial reservoirs influences: 1055
3. Consistency in catchment areas: 1031
4. Streamflow quality inspection: 654

About "Streamflow quality inspection", we visually checked the time series, while also checking information provided by the producers, e.g.: quality flag on streamflow times series (available for each time step), metadata about the influence, and the comments about influence (if missing data "influence" metadata, and also about the quality of the hydrometric mesures).

Note that as the producers update in real time the data (streamflow time series, quality flags, metadata, etc.), there can be some discrepancies from one extraction to another.

We think that this complementary information answers Mr. Janssen's questions. Since we want to keep the article content valid for subsequent version of the database, we would prefer not to include this information in the revised version of the manuscript.

---

## Author Comment (AC3)

**RC2 answer**

We thank Dr Larisa Tarasova for her feedback on the paper.

The submitted manuscript provides a description of catchment attributes and hydrometeorological time series for 654 French catchments. All the data described very clearly, manuscript is well-organized and well-written. The visualization tool is amazing! CAMELS-FR will be a very valuable addition for the community of large sample hydrology. I have just a few editorial suggestions.

**Editorial comments**

**Line 45-48**: It would be helpful to specify here a possible nature of these outliers (apart from karstic catchments).

Beyond karstic catchments, we also have groundwater-dominated catchments, fed by the chalk aquifer of the Parisian Basin, e.g. codes H504063010 ("Le Cailly à Notre-Dame-de-Bondeville"), E645651001 ("La Nièvre à l'Étoile"), or E540031001 ("La Canche à Brimeux"), they all have very high baseflow indices.
This information will be added in the revised version.

**Figure 2:** This is not critical, but would look nicer if all maps would have the same style in terms of displaying administrative boundaries and areas outside France. Consider homogenizing the panels of this Figure.

Thank you for your suggestion, we will update the figure adding administrative boundaries on all maps.

**Line 162:** in some stations
**Line 169:** lack of artificial reservoirs
**Line 170:** sufficient streamflow quality

Thank you for correcting the sentences to be more readable, we will follow your recommendations.

**Table 1:** explain what SIM2 stands for

Thank you for pointing out. In Sect 5.1, line 219 we mention that SIM2 stands for **S**AFRAN-**I**SBA-**M**ODCOU hydrometeorological model chain, which is maintained by Météo-France, and which is used to produce operationally the SAFRAN interpolated product

**Table 2:** Which attribute is meant here by "Other". I did not find a corresponding explanation in Section 6. Please clarify.

The variable that corresponds to the "Other" class is related to the site metadata variable "sit_publication_rights". In our dataset, all the stations have this variable filled with "Publication for the general public", and we think that on the public version of the Hydroportail website this variable will always have this value. There is another version of the Hydroportail website for the producers where they have some additional sites and stations that are not public. We decided to keep all the

metadata available in Hydroportail in the CAMELS-FR dataset and as in the future, there may be other metadata that may not be classified among the 9 main classes presented on Table 2, we decided to classify this specific variable under a more generic class name such as "Other". We did not want to specify this variable to a "License" class to avoid confusion about the general license of our dataset.

**Figure 7:** Very nice!

Thank you very much.

---

## Author Response (AR2)

**Response to editor**

**ESSD revision**

*CAMELS-FR dataset: A large-sample hydroclimatic dataset for France to explore hydrological diversity and support model benchmarking*

Editor's comments are colored in green.

**Editor answer**

**Dear authors of the CAMELS-FR data paper,**

**thank you for revising the manuscript and taking the referees' comments into account. As the (relatively minor) comments seem to be addressed satisfactorily, I will not send it out for another round of review again. However, the community comment by Joseph Janssen describes a valid point. It would indeed be good to provide the information about which criterion filtered out how many of the initially possible catchments for the dataset.**

**As the paper first and foremost refers to exactly this version of the dataset, you could give the numbers for this version and mention in the text that there will be updates of the dataset on the repository and the numbers might change in future versions.**

**Please revise the paper accordingly to also address the community comment.**

Dear editor, thank you for your feedback. To address Joseph Janssen's community comment, we added information regarding the number of catchments selected at each stage of the criteria filters. We have slightly modified the text to add this information:

**4 Selection of the catchment set**

The second step to build the `CAMELS-FR dataset` was to select the catchments based on the following criteria: (i) hydrometric time series availability (Sect. 4.1), (ii) limited artificial reservoirs influences (Sect. 4.2), (iii) consistency in catchment

175 areas (Sect. 4.3), (iv) sufficient streamflow quality (Sect. 4.4).  Starting from 4667 catchments, the successive application of these four selection criteria resulted in sub-sets of 1313, 1055, 1031 and 654 catchments respectively.

**4.1 Selection on hydrometric time series availability**

180 We excluded catchments with less than 30 years of complete data over the 1970-2021 period. This record length was arbitrarily chosen because it allows robust statistical analyses. A year was considered complete when it had less than 20 % of missing data.

**4.2 Artificial reservoirs influences**